# Modeling of Nanofiltration Process Using DSPM-DE Model for Purification of Amine Solution

**DOI:** 10.3390/membranes11040230

**Published:** 2021-03-24

**Authors:** Asma Ghorbani, Behrouz Bayati, Enrico Drioli, Francesca Macedonio, Tavan Kikhavani, Mirko Frappa

**Affiliations:** 1Department of Chemical Engineering, Ilam University, Ilam 69315-516, Iran; asmaghorbani94@gmail.com (A.G.); t.kikhavani@ilam.ac.ir (T.K.); 2Institute on Membrane Technology, ITM-CNR, c/o University of Calabria, via P. Bucci, 17/C, 87036 Rende, Cosenza, Italy; e.drioli@itm.cnr.it (E.D.); f.macedonio@itm.cnr.it (F.M.); m.frapa@itm.cnr.it (M.F.); 3Department of Environmental and Chemical Engineering, University of Calabria, via P. Bucci 45/A, 87036 Rende, Cosenza, Italy

**Keywords:** gas purification, amine solution, heat stable salts, feed flow, nanofiltration, DSPM-DE model

## Abstract

The formation of heat stable salts (HSS) during the natural gas sweetening process by amine solvent causes many problems such as corrosion, foaming, capacity reduction, and amine loss. A modeling study was carried out for the removal of HSS ions from amine solution using nanofiltration (NF) membrane process that ensures the reuse of amine solution for gas sweetening. This model studies the physics of the nanofiltration process by adjusting and investigating pore radius, the effects of membrane charge, and other membrane characteristics. In this paper, the performance of the ternary ions was investigated during the removal process from methyl di-ethanol amine solution by the nanofiltration membrane process. Correlation between feed concentration and permeate concentration, using experimental results with mathematical correlation as C_i,p_ = f (C_i,f_) was used in modeling. The results showed that the calculated data from the model provided a good agreement with experimental results (R^2^ = 0.90–0.75). Also, the effect of operating conditions (including feed pressure and feed flow rate on ions rejection and recovery ratio across the flat-sheet membrane) was studied. The results showed that the recovery and rejection ratios of the NF membrane depend on the driving pressure across the membrane. While the driving pressure is affected by the feed flow conditions and feed pressure.

## 1. Introduction

Natural gas is a nonrenewable energy source that usually contains undesirable spe-cies such as hydrogen sulfide (H_2_S), and carbon dioxide (CO_2_), which as well as being harmful to human health cause problems such as corrosion, plugging, freezing, erosion, and environmental hazards [1,2,3,4,5,6]. Therefore, the removal of undesirable species is an important part of natural gas sweetening industries. Usually, this is done using an amine-based solvent such as methyl diethanolamine (MDEA) [7,8,9,10,11,12]. The reaction between MDEA, H_2_S, and CO_2_ forms heat stable salts (HSS) that cannot be regenerated through heating. The maximum concentrtions of the dominant anionic components in amine solvents are as follows: sulfate—500 ppm; acetate—1000 ppm; glycolate—500 ppm; oxalate—250 ppm; formate—500 ppm; chloride—500 ppm respectively [13,14]. The accumulation of these salts in amine solution leads to a reduction in the efficiency of the CO_2_ absorption process and operational problems such as corrosion, fouling, foaming, high viscosity, capacity reduction [15,16,17,18]. Therefore, the removal of HSS from the amine solvent is an indispensable part of gas sweetening industries. Various technologies that can be used to remove HSS ions from amine solvent include electrodialysis, thermal reclamation, ion exchange and distillation [19,20,21,22]. 

Another effective technique for the removal of HSS from amine is nanofiltration (NF). NF is a pressure-driven membrane operation that has properties between ultrafiltration (UF) and reverse osmosis (RO) [23,24,25,26,27]. NF has a lower operating pressure and higher permeate fluxes than RO and higher rejection than ultrafiltration [26,28,29,30,31,32]. The ion rejection in the NF membrane is mainly based on size (stearic), dielectric exclusion, and charge (Donnan exclusion) mechanisms [33,34,35,36,37,38,39,40,41,42,43,44]. The removal of HSS ions as single salt from 30 wt. % MEA (Ethanolamine or 2-aminoethanol) solutions using NF (Koch MPF-34 and MPF-36) mem-branes and up to 30 bar was studied by Lim et al. [45]. They found that NF membranes can remove 80% of HSS ions, with MEA rejection less than 7%. In a study, we investigated the separation of a multicomponent mixture of HSS ions from the different concentrations of MDEA solution via NF membrane. The result showed that the separation efficiency of ions was close to 80% [46]. Also, in another work, we estimated transport parameters for a binary system of ions in amine solution through NF membrane [47]. We found that film theory and the extended Nernst-Planck model provide better agreement with experimental data. 

A comprehensive and practical method for study the mechanism of NF is the Don-nan Steric Pore Model with dielectric exclusion (DSPM-DE). This method investigates the Extended Nernst-Planck equation for each ion species through the NF membrane and in order to verify the steric exclusion effects, Donnan exclusion, and dielectric exclusion, the boundary conditions are used at the membrane surfaces [48,49]. The Extended Nernst-Planck equation involves a combination of hindered diffusion, hindered advection, and electro-migration of solutes [50]. Also, these equations have been successful in describing the nanofiltration performance compared to other existing results [51]. The DSPM-DE has been widely applied in NF systems for feed solutions with single-species and mixed electrolytes. However, the DSPM-DE model has some limitations and many basic assumptions. These include strong sensitivity to some experimental errors, the requirement of excessive experiments to determine necessary parameters, and questionable underlying assumptions of the mean-field approach. Also, although the DSPE-DE applies several significant ion transport mechanisms, it lacks the key features as an ideal model for performance prediction [52]. Chakraborty et al. [53] developed a transport model based on the Nernst-Planck equation for the removal of fluoride from contaminated groundwater by cross-flow NF. They found that their model could predict the performance of the system as reflected in relatively low relative error and high overall correlation coefficient. Roy et al. [48] presented a model based on DSPM-DE to evaluate the effect of various operating parameters on the ion rejection from water by using spiral-wound and flat-sheet modules. They obtained similar results for the same operating conditions and membranes in the spiral-wound and flat-sheet configurations. The NanoFiltran computer program was developed to simulate the mass transport of multi-ionic aqueous solution in charged NF mem-branes by Geraldes et al. [54], using the DSPM-DE model. The results were obtained by them showed that NanoFiltran is a useful tool for accurate prediction of the mass transfer of multi-ionic solutions in the NF membrane. The separation of solutes in the NF process according to the modified extended Nernst-Planck equation was investigated by Zerafat et al. [55]. They considered the activity coefficient variations due to variations in ionic strength. Their results indicated that the rejection is significantly lower than the predicted rejection with increasing the concentration. 

To the best of the authors’ knowledge, ion transport mechanisms from amine solutions by NF membranes have not been studied so far, and all works have focused on the transfer of ions in water. In this work, we provided an experimental study sustained by the equations of the DSPM-DE model to evaluate the removal of ions from amine solutions through the nano-filtration membrane. The study is based on the mass transfer applied on a flat-sheet element. It can also describe the behavior of an individual element for various feed flow rates, transmembrane pressures, and feed compositions.

## 2. Materials and Methods

### 2.1. Experimental Set-Up

NF experiments were performed at the laboratory-scale using a cell equipped with an NF-3 polyamide flat-sheet membrane (Sepro Membranes, Inc., Oceanside, CA, USA) operated in cross-flow (Figure 1) (as were conducted in the preveous study [46]). The effective area was 0.0113 mThe specifications provided by the manufacturer indicated that the NF-3 polymeric membrane consists of three layers, including the polyamide surface layer, and polyester support with a polysulfone substrate. The characteristics of the NF-3 membrane are provided in Table 1. The NF membrane was immersed in distilled water for 24 h prior to use. The feed pressure, temperature, and pH in all tests were fixed at 70 bar, 35 °C, and 10, respectively. The filtration operation was performed at a high pressure of 70 bar, due to the increase in osmotic pressure for two reasons: (1) the first layer of the membrane is a polyamide and hydrophobic. (2) the high concentration of amine (45 wt. % MDEA). The temperature was maintained using cold water circulation. The feed flow rate was 0.693 L/h in all experiments, and was controlled using the pressure pump and the inlet valve. In order to keep the feed concentration constant, the permeate samples were returned to the feed tank after each test.

The feed was a ternary mixture of C_2_H_3_NaO_2_, CH_2_O_2_, and Na_2_SO_4_ in 45 wt. % MDEA solution, which was purchased from Ghatran Shimi Tajhiz, Tehran, Iran. MDEA was supplied by Ilam Gas Treating Co., Ilam, Iran.

For rejection experiments, the concentration of ions in permeate was measured by an ion chromatography instrument (I.C.; metrosep a SUPP 5–250, Herisau, Switzerland). 

Table 1 reports the characteristics of the NF-3 membrane which were provided by manufacturers and mentioned in the literature.

### 2.2. Process Modeling

To calculate the concentration profile on the feed side was assumed:the gradient of the concentration is neglected in the width and length direction along feed side;flow rate profile along the module is obtained by total mass balance equation;mass transfer by diffusion in the axial direction is neglected due to the high flow rate of solvent.

The schematic diagram of the flat sheet membrane for the experimental tests is illustrated in Figure 2.

The mass balance on the feed side for “*i*” component and for the total system is as follows:

Component mass balance:(1)$$\frac{d\left(Q_{f}.C_{i,f}\right)}{dZ}=-J_{V}.P.C_{i,P}$$
(2)$$\frac{d\left(Q_{p}.C_{i,P}\right)}{dZ}=J_{V}.P.C_{i,P}$$
where,
(3)$$C_{i,P}=f\left(C_{i,f}\right)$$

Total mass balance:(4)$$\frac{dQ_{f}}{dZ}=-J_{V}.P$$
(5)$$\frac{dQ_{P}}{dZ}=J_{V}.P$$
where *Q_f_* and *Q_P_* are volumetric flow rates (L/h) in the feed side and the permeate side, respectively. Also, *C_i,f_* and *C_i,P_* are “*i*” component concentration (mg/L) on the feed and permeate sides, respectively, *J_V_* is permeating flux (L/h.m^2^) and *P* is perimeter (m^2^).

Boundary conditions:(6)$$@\;z=0\;:Q_{f}=Q_{f0},\;C_{i,f}=C_{i,f0}\;@\;z=0\;:Q_{P}=Q_{P0},\;C_{i,P}=C_{i,P0}$$

The concentration profile on the feed side was obtained by the numerical solution of Equations from (1) to (5) applying the boundary conditions (6). *C_i,P_* was calculated from the quadratic function which was obtained by fitting the experimental values of *C_i,P_* versus *C_i,f_*. The set of differential equations was solved by the 4th order Runge-Kutta method using MATLAB software (MATLAB R2014a, Available online: www.mathworks.com/products/matlab/ (accessed on 11 December 2020).

The static error metrics, including the average absolute relative error, AARE, and the coefficient of determination, R^2^, are applied to evaluate the validity of the results. AARE and R^2^ were determinate as follows [57,58,59,60]:(7)$$AARE=\frac{1}{n}\overset{n}{\underset{i=1}{\sum }}\left|\frac{R_{i}^{exp}-R_{i}^{calc}}{R_{i}^{exp}}\right|$$
(8)$$R^{2}=1-\frac{\sum_{i=1}^{n}{\left(R_{i}^{exp}-R_{i}^{calc}\right)}^{2}}{\sum_{i=1}^{n}{\left(R_{i}^{exp}-\bar{R^{exp}}\right)}^{2}}$$
where *R_i_^exp^*, *R_i_^calc^*, $\bar{R^{exp}}$, n are the rejection of the species “*i*” for the experimental data, the rejection predicted for the species “*i*”, the average value of the rejection and the number of total data, respectively.

According to the one-dimensional continuity equation, the ion molar flux (*J_i,pore_*) is independent of ions position within pores in steady-state and is given by [48,54]:(9)$$J_{i.pore}=C_{i.P}J_{V}$$

The observed rejection (*R_obs_*), intrinsic rejection (*R_int_*) and permeate flux (*J_V_*) were calculated by the following equations [41,61]:(10)$$R_{obs}\left(\%\right)=\left(1-\frac{C_{i,p}}{C_{i,f}}\right)\times 100$$
(11)$$R_{int}\left(\%\right)=\left(1-\frac{C_{i,p}}{C_{i,m}}\right)\times 100$$
(12)$$J_{V}=\frac{V}{A.t}$$
where *A* (m^2^), *V* (L) and *t* (h) are the effective membrane area, permeate volume, and time, respectively. C_i,m_ is concentrations at the membrane surface for “*i*” component, which can be calculated with the concentration polarization equation.

If the mass transfer coefficient was not infinite [54], concentration polarization at the feed-solution/membrane interface is given by the below equation [61,62]:(13)$$\frac{C_{i,m}-C_{i,f}}{C_{i,f}-C_{i,P}}=\exp\left(\frac{J_{V}}{k_{i}}\right)$$
where *k* is the mass transfer coefficient for laminar flow in the feed channel obtained from [63,64]:(14)$$Sh_{i}=1.85\;{\left(ReSC_{i}d_{h}/L\right)}^{1/3}$$
(15)$$Re=\frac{\rho ud_{h}}{\mu },\;Sc_{i}=\frac{\mu }{\rho D_{i,\infty }},\;Sh_{i}=\frac{k_{i}d_{h}}{D_{i,\infty }}$$
where *d_h_* is the hydraulic diameter of feed channel, *u* is the bulk velocity of flow, *L* is the channel length, and *D_i,∞_* is the diffusion coefficient of ion “*i*”. The diffusion coefficient of ions in the electrolyte solution was calculated by molecular simulation software.

Also, the hydraulic pressure drop (Δ*P_loss_*) along the feed flow direction is given by the following equation,
(16)$$\Delta P_{loss}=-\frac{f}{2}\frac{L}{d_{h}}\rho u^{2}$$
where *L* is the length of the feed channel along the feed flow direction and f the friction factor given by
(17)$$f=\frac{6.23}{Re^{0.3}}$$

The trans-membrane solvent flux is given by the following equation,
(18)$$J_{V}=\left(\left(P_{f}-P_{P}\right)-\Delta \pi \right)\left(\frac{r_{Pore}^{2}}{8\mu \left(\frac{\Delta x}{A_{k}}\right)}\right)$$
where *P_f_* and *P_P_* and ∆*π* are pressure in feed bulk, permeate pressure, and osmotic pressure, respectively. *μ* is solvent viscosity. *r_pore_*, ∆*x*, and *A_k_* are pore radius of the membrane, membrane active layer thickness, and porosity of membrane, respectively.

The DSPE-DE was used to describe the transfer of ions through pores under concentration gradient, inertia forces, and electric field using the extended Nernst-Planck equation [48,54,61]. These equations are summarized in Appendix A.

The Stokes radius were obtained by the Stokes-Einstein correlation [65,66]:(19)$$r_{s}=\frac{k_{B}T}{6\pi \mu D_{i,\infty }}$$

In the present study, the feed and permeate channels were considered as thin rectangular ducts, with heights of 0.7 mm and 0.3 mm, respectively. Moreover, each membrane sheet has a dimension 1 m × 1 m.

## 3. Results and Discussion

### 3.1. Correlation for C_i,P_ = f(C_i,f_)

Feed concentration has a significant effect on nanofiltration performance. The variation of ion concentrations in the feed leads to a change of ion concentrations in permeate. For this purpose, the ion concentrations in permeate can be expressed as a function of the concentration of ions in the feed (*C_i,P_* = *f*(*C_i,f_*)). Therefore, the effect of the feed concentration on the permeate concentration in 45 wt. % MDEA solution by NF-3 membrane for ternary salts was investigated at 70 bar and pH = 10, and the results were illustrated in Figure 3 Also, the correlation of permeate concentration and feed concentration in the amine solution was obtained as a quadratic function for each ion by curve fitting *C_i,P_* versus *C_i,f_* using experimental results in Figure 3. Since the NF-3 membrane has a negative charge, solutions with a pH of 10 lead to a strong negative charge of the membrane surface. Moreover, solutions with pH > 10 create scaling and fouling problems. On the other hand, the osmotic pressure of 45% wt. MDEA solution was high according to the Van’t Hoff equation (π = CRT, where π is the osmotic pressure, R is constant of proportionality also called general solution constant or gas constant, C is the concentration of the solution and T is the temperature [67]), and the rejection of MDEA by NF-3 was 1.2% at 70 bar. For the reasons above illustrated, the tests were performed at 70 bar and pH = As can be seen, the permeate concentration increased with increasing feed concentrations for all ions. This is due to the fact that in NF and RO, the solute flux is described by $J_{s}=B.\Delta c_{s}$ where B is the solute permeability coefficient. Therefore, the higher concentration of ions in the feed will lead to the lower quality of the permeate (since the solute leakage through the membrane is directly proportional to the solute concentration at the membrane feed side surface). Also, with the increase in the concentration of ions, the fixed negative charge on the membrane surface was partially neutralized by the counter ions leading to a decrease in the electrostatic repulsion between the ions and the membranes [53]. Hence, the concentration of ions in the permeate increases with increasing feed concentrations. On the other hand, the effect of concentration polarization (the ions accumulation in the boundary layer) can increase the concentration of ions in the permeate and leads to a reduction in rejection. The increase of the concentration polarization by increasing the feed concentration can be better investigated by evaluating the observed and intrinsic rejection, which were calculated using Equations (10)–(15) and plotted as a function of the feed concentration in Figure 3. As Figure 4 shows, the ion rejection decreased with increasing the ion concentration at the same operating pressure. Moreover, the difference between the observed rejection and intrinsic rejection increases when the concentration of ions rises. These indicate that the concentration polarization layer on the membrane surface increases when the ion concentration enhances [68].

The transport of ions through the NF membrane was described by the extended Nernst–Planck equation. This equation includes three transport mechanisms; diffusion (due to concentration gradient), Convection (due to pressure gradient) and Electromigration (electric potential gradient) [62,69,70].

The analysis of the transport mechanisms indicates that diffusion transport for all of the ions has a more contribution due to the concentration polarization in the system (as discussed earlier) and the concentration gradient. The contribution of electromigration for sulfate is lower than formate and acetate because sulfate is a divalent ion and hence the sulfate repulsion exerted by the NF membrane is higher than other anions. On the other hand, convection contribution to the transport of ions is almost constant because applied pressure is constant, and the pressure gradient does not vary. Also, due to higher negative charge and size, divalent ions (sulfate) are more likely to be rejected than monovalent ions (formate and acetate) due to the Donnan, dielectric and steric interactions.

### 3.2. DSPM-DE Validation on Experimental Data

Validation of results obtained was performed by comparing the experimental tests with the mathematical equations reported at various concentrations and the results are shown in Figure 5. 

The empirical relations between the permeate concentration and the feed concentration were used to predict the theoretical values. In this way, it is possible to predict the ion concentration changes along the feed direction at the feed side and permeate side. In this work, due to the small dimensions of the membrane, the ion rejection in the output was calculated by the permeate concentration integral on the surface and compared with the experimental results. The comparison was conducted at a feed flow rate of 0.693 L/h. The results showed that the data provided a good agreement with experimental results (AARE ≈ 2–10% and R^2^ ≈ 0.90–0.75). On the other hand, these correlations were exploited as the first guesses to compute the DSPM–DE model’s data in Equations (T1)–(T10). The parameter of the model and physical properties are mentioned in Table 2. The results of the model are shown in Figure 6 and are compared with experimental results. A glance at Figure 6 reveals that there is acceptable compatibility between the results of the model and the experiments. 

To the best of the authors’ knowledge, ion concentrations in amine solutions at different points between the feed side, the membrane surface, and the permeate side by the NF membrane have not been reported in the literature and a comparison could not be done. 

### 3.3. Ion Diffusion Coefficients Calculation

The diffusion coefficients of ions in a 45 wt. % MDEA electrolyte solution were estimated using the molecular dynamics simulation software. The physical properties of ions used in these calculations were reported in Table 2. Considering the importance of calculating the mass transfer coefficient of ions for the investigation of concentration polarization, mass transfer coefficients of ions were calculated by Equations (14) and (15) in different feed flow rates and are shown in Figure 7. As can be seen, with increasing feed flow rate, the mass transfer coefficient of ions increases due to the increase in the Reynolds number in the channel. The concentration boundary layer is thinner at greater Reynolds numbers, leading to a decrease in the concentration polarization. For the rectangular channel without spacers, the transition Reynolds number value from laminar to turbulent occurs at larger than 1000 and in the presence of spacers is 150 < Re < 300 [72]. In this work, the Reynolds number at the inlet for the maximum feed flow rate of 1000 L/h is 118, which indicates the laminar flow.

### 3.4. Analysis and Model Description

In this section, the effect of various parameters including feed flow rate and feed pressure on the nanofiltration membrane performance for the removal of ions from the amine solution was investigated. The simulations were carried out for ternary salts of formate, acetate, and sulfate with concentrations of 300, 200, and 150 mg/L, respectively, at 45 wt. % MDEA solution.

#### 3.4.1. Effect of Feed Flow Rate on Feed Pressure Variation along the Feed Flow Direction

The pressure distributions along the feed flow direction for different feed pressures are given in Figure 8. As can be seen, the feed pressure along the membrane decreased for higher feed flow rates due to higher Reynolds numbers and greater hydraulic losses. On the other hand, the hydraulic loss on the permeate side is negligible, due to the very small Reynolds number on the permeate side, thus the hydraulic pressure on the permeate side along the flow direction remains constant. Therefore, hydraulic pressure is mainly a function of feed pressure [48].

#### 3.4.2. Effect of Feed Flow Rate on Ions Concentration along Flow Direction

The concentration distribution of ions at the feed bulk and on the membrane surface along membrane length at different flow rates are illustrated in Figure 9 and Figure 10a–c. As can be seen, the concentration of ions at feed bulk and on the feed membrane surface increased with decreasing feed flow rates for all three ions. This can be due to the reduction of the Reynolds number along the feed channel as a result of the decrease in feed flow rate along the membrane [48]. Moreover, this is due to the drop in feed pressure caused by the hydraulic losses along the feed flow direction as shown in Figure 7. Furthermore, by increasing the Reynolds number and thus the mass transfer coefficient of ions (Figure 7), the polarization concentration becomes smaller. Therefore, a higher feed flow rate minimizes the effect of concentration polarization by sweeping the ions from the membrane surface [60,73,74]. Conversely, with decreasing the feed flow rate, the mass transfer coefficient decreases and the concentration polarization increases on the membrane surface. Thus, the concentration of ions on the membrane surface is greater than in the feed bulk, which results in the formation of a boundary layer. On the other hand, as indicated in Figure 11a–c, with decreasing feed flow rate, the concentration of ions increased on the permeate side, which was resulted from an increase in the concentration polarization in the feed side and a reduction in the rejection of ions. At the lowest flow rate of the feed, due to the accumulation of solute on the membrane surface and an increase in concentration polarization at the feed side, the transfer of solutes to the permeate channel increases [48].

#### 3.4.3. Effect of Inlet Feed Pressure on the Ions Concentration along Flow Direction

Figure 12 shows the concentration distribution of ions along flow direction with increasing inlet feed pressure. As shown in Figure 12a–c, the concentration of ions in the feed bulk increased with increasing inlet feed pressure for all ions. The difference between the transmembrane osmotic pressure difference and the applied pressure is the only driving force for solvent transport in both the solution-diffusion and the DSPM-DE models [50]. Thus, with increasing feed pressure, the driving force across the membrane for solvent flux increases, but an increase in solvent flux with increasing inlet feed pressure does not result in an increased solute flux. This is due to the increase of the concentration polarization with increasing feed pressure owing to the accumulation of ions at the feed side. Therefore, with increasing pressure along the flow direction, the concentration of ions at the feed side increases.

#### 3.4.4. Effect of Inlet Feed Pressure and Feed Flow Rate on Rejection and Recovery Ratios

The ion rejection rates of the membrane at different feed flow rates and inlet feed pressures are shown in Figure 13. As shown, the rejection ratio for all ions at different pressures is almost constant, and a slight reduction in rejection at higher pressures can be attributed to an increase in the concentration polarization with increasing feed pressure. The rejection ratio for all ions increases with an increase in the feed flow rate, but at the feed flow above 500 L/h, it reaches an almost constant value. This can be because, initially, an increase in driving force improves solvent transfer by increasing the flow rate, but with further increasing of the flow rate, the driving force is reduced due to the effect of hydraulic pressure drop (Figure 8), resulting in lower solvent flux and thus lower rejection ratio [48].

Figure 14 shows the recovery ratio for different feed pressures and flow rates. It is illustrated that the recovery ratio increased with an increase in the inlet feed pressure, and this can be due to higher solvent flux through the membrane with an increase in the driving force of the process. On the other hand, the recovery ratio decreased with increasing feed flow rate. The recovery ratio was obtained from the ratio of the permeate flow rate at the outlet to the inlet feed flow rate. Because the osmotic pressure of the amine solution is significantly high and its osmotic pressure dominates (the osmotic pressure of the ions is negligible); therefore, the net driving pressure (ΔP − Δπ) will be a constant value. Hence, the trans-membrane flux of solution (Equation (18)) will be constant due to the fixed membrane properties and constant Driving Pressure. Since the flux in the permeate side is caused by flux coming from the feed side, therefore, the permeate flow will remain constant at various feed flows, as a consequence, the recovery ratio reduced by an increase in feed flow.

## 4. Conclusions

This work is a comprehensive experimentation that allows to obtain general trends and to describe a criterion for the experimental procedure required for a complete characterization of NF membrane performances in the separation of HSS ions from amine solutions. In the experimental section, the effect of ion concentrations in the feed side on the rejection of ternary salts from 45 wt. % MDEA solution was investigated by NF falt-sheet cross-flow process. The rejection of ions from the MDEA solution was decreased with an increase in ion concentrations.

In this study, nanofiltration tests have been developed by mass transfer balance on the flat-sheet module. The effects of different parameters including feed pressure and flow rate on the rejection of ions and the recovery ratio of the NF membrane for the separation of ternary salts from 45 wt. % MDEA solution have been studied. The feed concentration, permeate concentration, and concentration at the membrane surface along flow direction have also been provided at different feed flow rates and feed pressure. The permeate concentrations were expressed as a function of feed concentrations using experimental results for each ion in the modeling study. The obtained data was examined by comparing the DSPM-DE equations with experimental data to remove the ions of 45 wt. % MDEA solution. A good agreement was observed to predict the performance of nanofiltration processes on a large scale and for a wide range of operating conditions. This work indicates that the performance of NF membranes for the removal of ions from MDEA solution can be well predicted by the DSPM-DE modal. This model can be used as a reliable method to simulate the performance of NF in large-scale systems and for a variety of aqueous solutions in a wide range of operating conditions and reduce dependence on experimental data.

The results obtained showed that the concentration of ions in the feed side and in permeate side increased with increasing pressure and/or decreasing the feed flow rate along the flow direction. The results have shown that the rejection ratio of the NF membrane depends on concentration polarization and the driving pressure across the membrane. The recovery ratio increased with an increase in the inlet feed pressure due to the increase of solvent flux through the membrane.

## Figures and Tables

**Figure 1 membranes-11-00230-f001:**
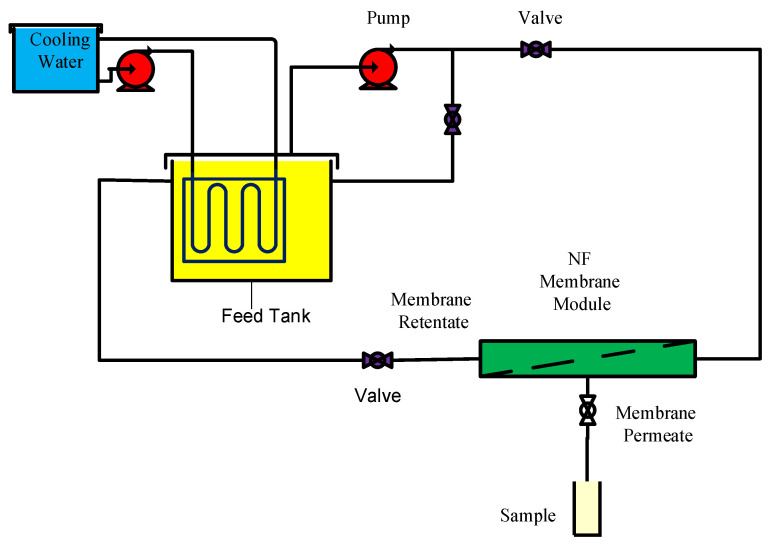
Schematic diagram for the nanofiltration (NF) membrane set-up.

**Figure 2 membranes-11-00230-f002:**
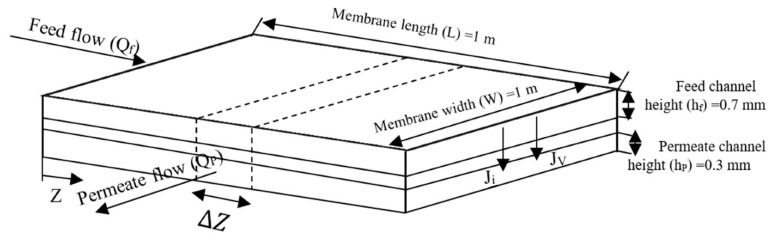
Schematic diagram of flat-sheet membrane.

**Figure 3 membranes-11-00230-f003:**
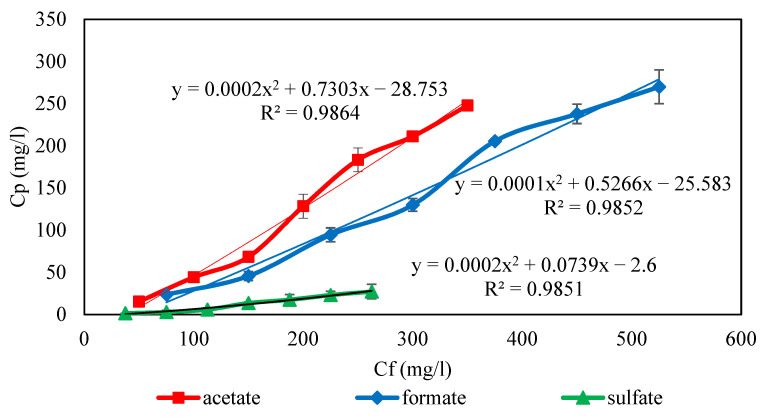
Effect of feed concentration on permeate concentration for ternary salts of 45 wt. % MDEA solution by NF membrane at 70 bar (T = 35 °C, pH = 10).

**Figure 4 membranes-11-00230-f004:**
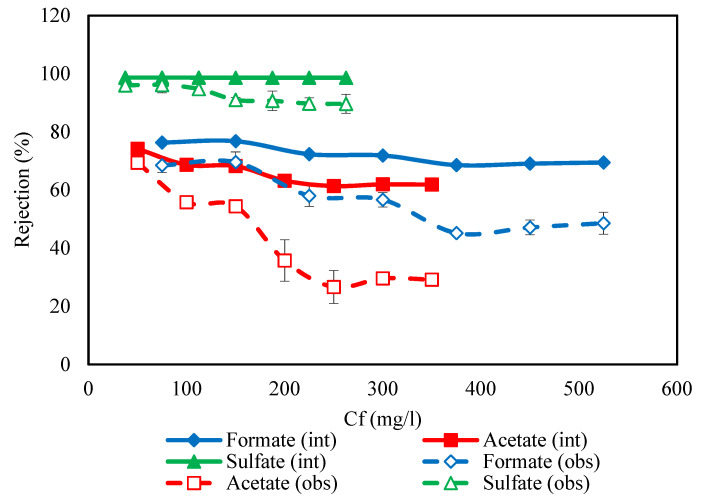
Comparison of observed and intrinsic rejection for acetate, formate, and sulfate in the NF-3 membrane at 35 °C and 70 bar.

**Figure 5 membranes-11-00230-f005:**
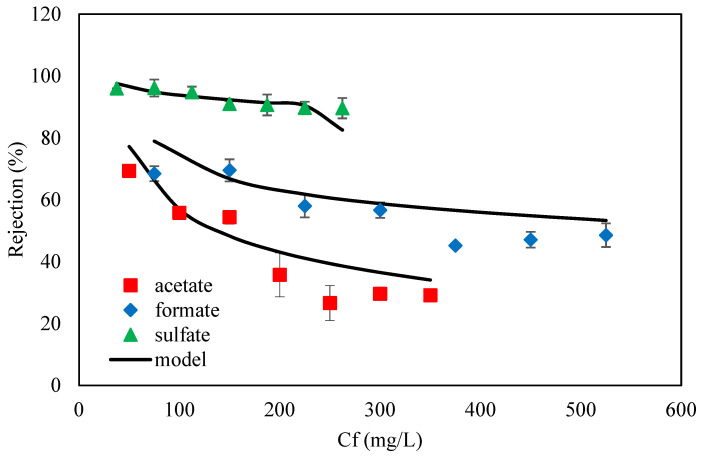
Comparison of the experimental data with the obtained results of the mathematical equations reported for acetate, formate, and sulfate in the NF-3 membrane at 35 °C and 70 bar.

**Figure 6 membranes-11-00230-f006:**
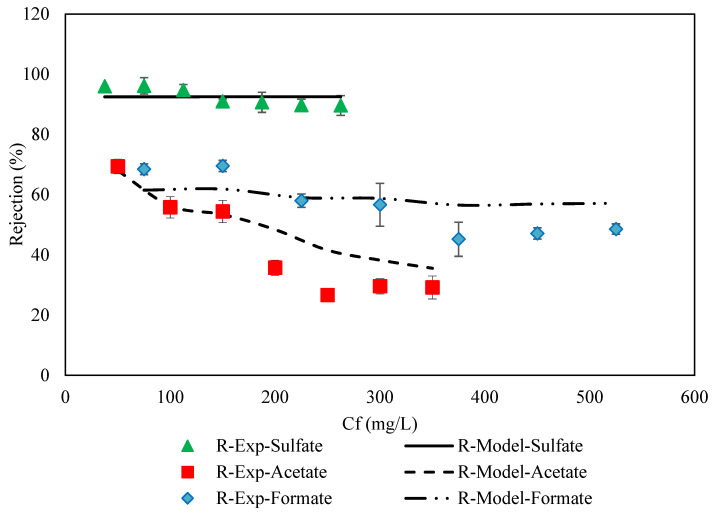
Comparison between experimental data and model results for the ternary salts in 45 wt. % MDEA solution by NF membrane at 70 bar, 35 °C, pH = 10 and feed flow rate of 0.693 L/h.

**Figure 7 membranes-11-00230-f007:**
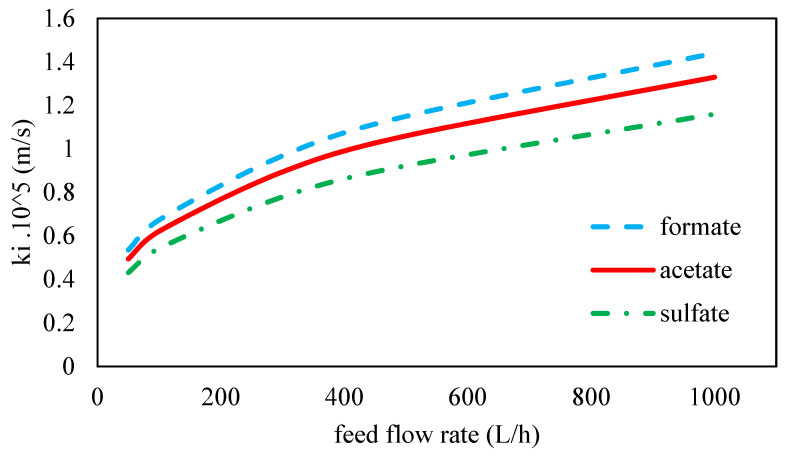
The mass transfer coefficient as a function of feed flow rate for different ions in 45 wt. % MDEA solution at 70 bar and 35 °C.

**Figure 8 membranes-11-00230-f008:**
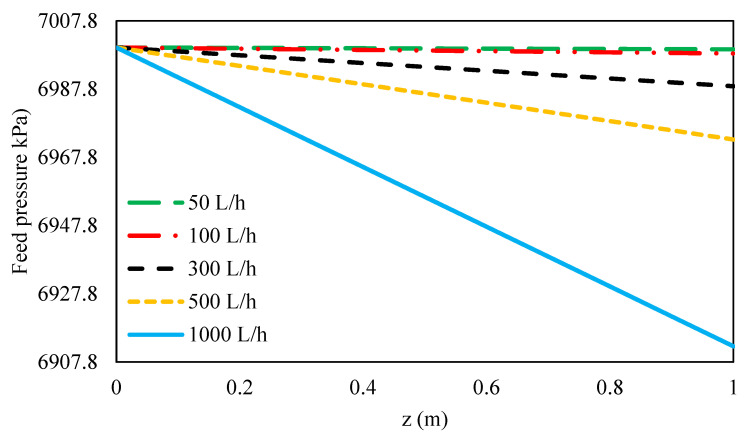
Feed pressure along the feed flow direction at different feed flow rates and 70 bar inlet feed pressure.

**Figure 9 membranes-11-00230-f009:**
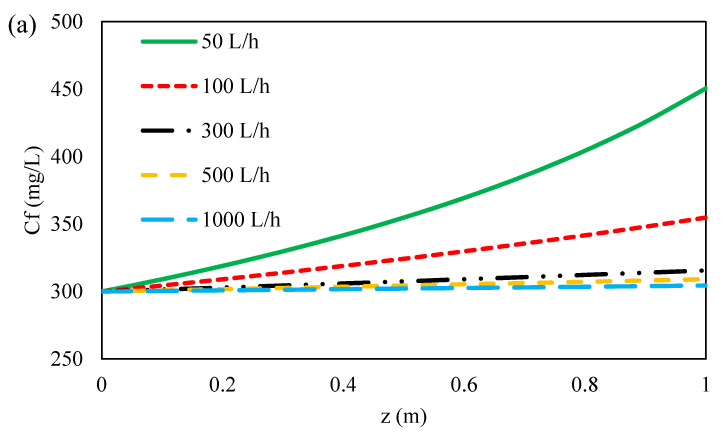
Ions concentration on the feed side for (**a**) formate, (**b**) acetate, (**c**) sulfate of 45 wt. % MDEA solution at different feed flow rates and 70 bar inlet feed pressure.

**Figure 10 membranes-11-00230-f010:**
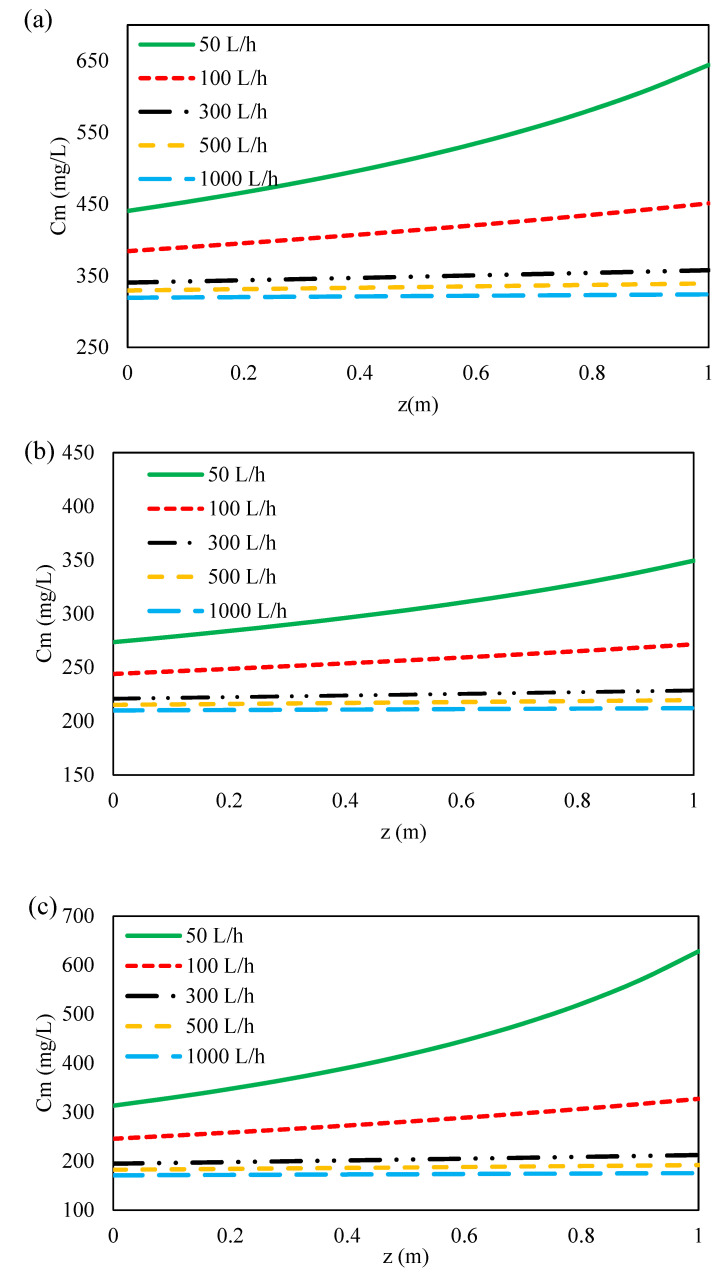
Feed concentration profile on membrane surface along the feed flow direction for (**a**) formate, (**b**) acetate, (**c**) sulfate at 70 bar inlet feed pressure and different feed flow rates.

**Figure 11 membranes-11-00230-f011:**
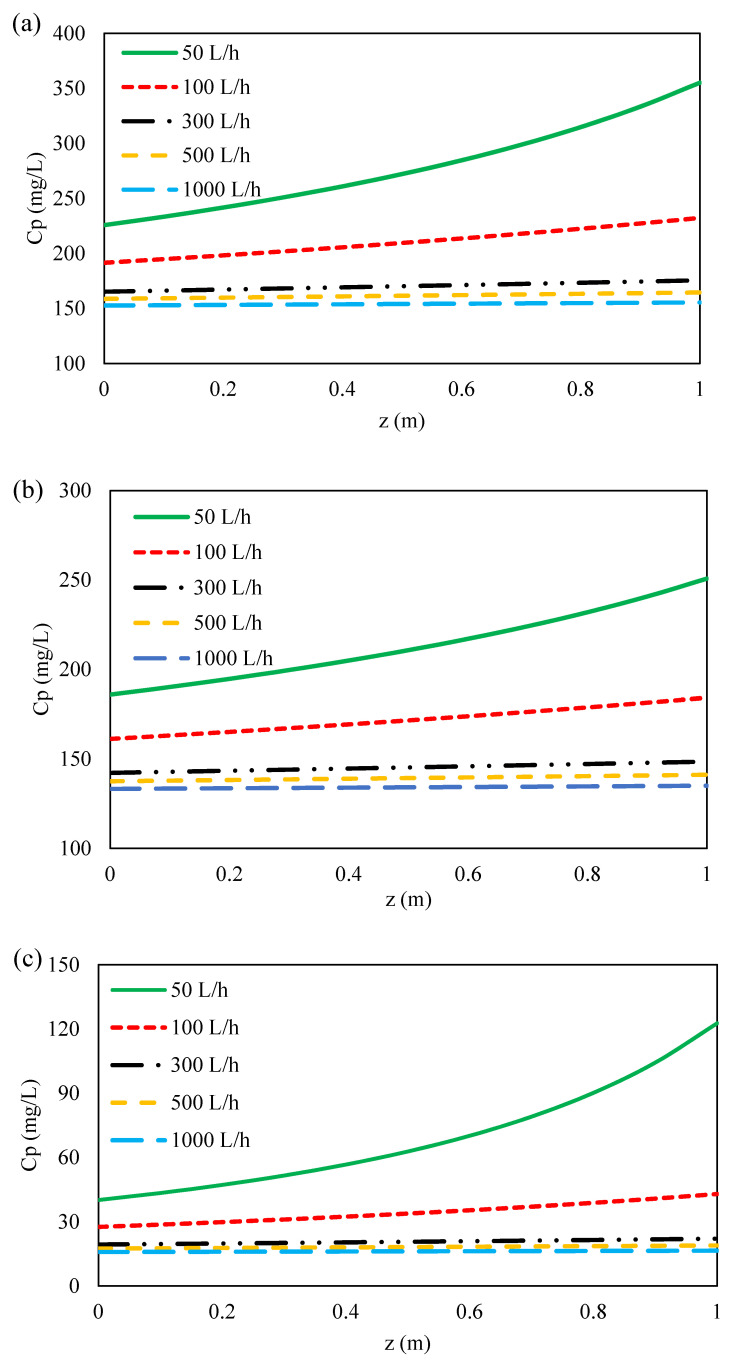
Permeate concentration variations along the feed flow direction for (**a**) formate, (**b**) acetate, (**c**) sulfate at different feed flow rates and 70 bar inlet feed pressure.

**Figure 12 membranes-11-00230-f012:**
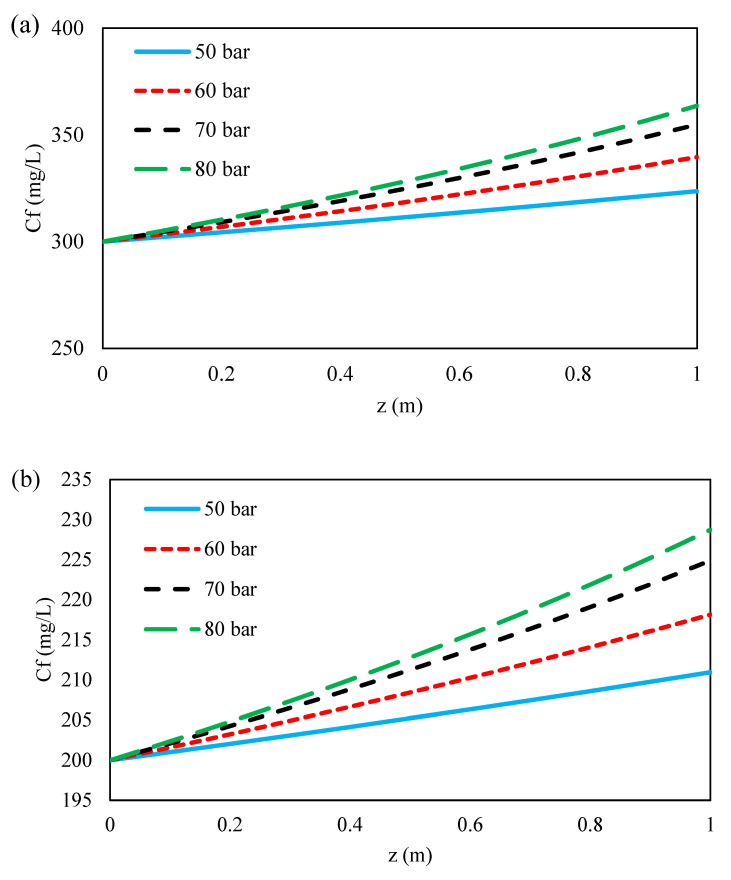
Feed bulk concentration profile during feed side for (**a**) formate, (**b**) acetate, (**c**) sulfate of 45 wt. % MDEA solution at 100 L/h feed flow rate and different inlet feed pressure.

**Figure 13 membranes-11-00230-f013:**
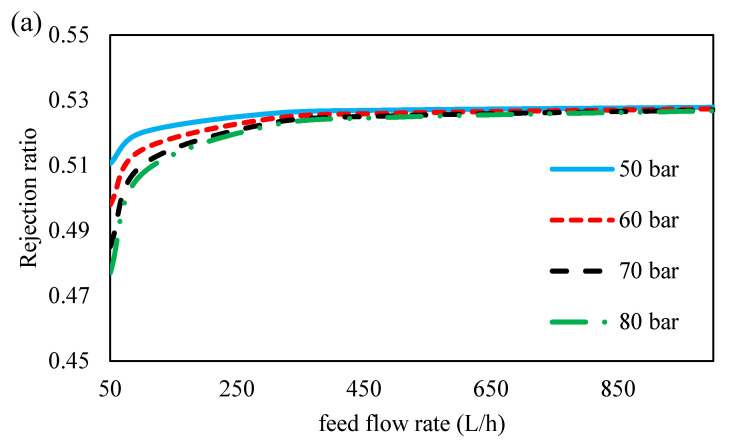
Rejection rate for (**a**) formate, (**b**) acetate, (**c**) sulfat, of 45 wt. % MDEA solution at different inlet feed pressures and feed flow rates.

**Figure 14 membranes-11-00230-f014:**
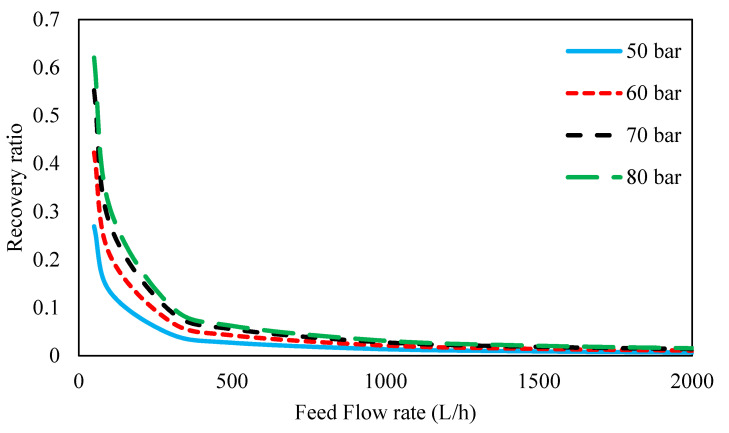
Recovery ratio for 45 wt. % MDEA solution at different inlet feed pressures and feed flow rates.

**Table 1 membranes-11-00230-t001:** Characteristics of nanofiltration membrane [56].

NF-3 Membrane						
MWCO (Da)	Pore Radius (nm)	Membrane Thickness (μm)	(∆x/A_k_) (μm)	Pure Water Permeate (L m^−2^ h^−1^ bar^−1^)	Operation Limits	Rejection (%)
250–300	0.55	0.9	0.51	8.86	50 °C, 83 bar, 3–10 pH	NaCl: 60% MgSO_4_: 98%

**Table 2 membranes-11-00230-t002:** Model parameters and physical properties of ions at 35 °C.

Parameters	Values	References
Feed flow rate (L h^−1^)	0.693	
Cross flow velocity (m s^−1^ 1 × 10^6^)	7.67	
Temperature maintained in units (K)	308	
Solute radius of HCO_2_^−^ ion (r_s_ nm)	0.00738	
Solute radius of C_2_H_3_O_2_^−^ ion (r_s_ nm)	0.00832	
Solute radius of SO_4_^2−^ ion (r_s_ nm)	0.0102	
Solute radius of Na^+^ ion (r_s_ nm)	0.116	[53]
Solute radius of H^+^ (r_s_ nm)	0.025	[56]
Bulk diffusivity of HCO_2_^−^ ion (D_i,∞_ × 10^9^ m^2^/s)	1.33	
Bulk diffusivity of C_2_H_3_O_2_^−^ ion (D_i,∞_ × 10^9^ m^2^/s)	1.18	
Bulk diffusivity of SO_4_^2−^ ion (D_i,∞_ × 10^9^ m^2^/s)	0.96	
Bulk diffusivity of Na^+^ ion (D_i,∞_ × 10^9^ m^2^ s^−1^)	1.9	[53]
Bulk diffusivity of H^+^ ion (D_i,∞_ × 10^9^ m^2^ s^−1^)	9.3	[56]
Mass transfer coefficient of HCO_2_^−^ ion (m s^−1^ × 10^5^)	1.44	
Mass transfer coefficient of C_2_H_3_O_2_^−^ ion (m s^−1^ × 10^5^)	1.33	
Mass transfer coefficient of SO_4_^2−^ ion (m s^−1^ × 10^5^)	1.16	
Boltzmann constant (k) (J K^−1^ × 10^−25^)	1.38066	[71]
Faraday constant	96,500	[71]

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
