# Peer review of "Modeling of Nanofiltration Process Using DSPM-DE Model for Purification of Amine Solution"

_membranes, 2021, doi:10.3390/membranes11040230_

Round 1

Reviewer 1 Report

The purpose of this study is to establish a model to evaluate the performance of ions in removing heat stable salts from amine. First of all, I should admit that the research topic of this article is very meaningful, but I also think that the presentation of the content in the article is very confusing. So, the paper needs reformulation in order to be clearer and to present a scientific format. Among them, the problems are mainly focused on the introduction of research topics and the description of the Figures & Tables. Here are my specific comments below:

Text content:

  1. The abstract is not well-written thus should be polished.
  2. Advantages and disadvantages of the proposed model in this field should be better discussed aimed to justify the applied structure.
  3. The content of each part of the article is more like a splicing and explanation of the experiment, without the slightest logic in the middle.

Figures & Tables:

  1. There are too many figures in this article and the typesetting is too messy, which makes it very troublesome to read. I hope that it can be deleted and typeset again.
  2. Please keep the font of the figures’ axis title consistent.
  3. The legend in the Fig. 3 is missing.
  4. Line 289: if there are no special circumstances, please keep the same font before and after.
  5. 11 needs to be redrawn.

I am sorry to say that this article is difficult to read. The author needs to rethink how this article should be expressed clearly, and the current version does not meet the requirements of publication.

Author Response

The purpose of this study is to establish a model to evaluate the performance of ions in removing heat stable salts from amine. First of all, I should admit that the research topic of this article is very meaningful, but I also think that the presentation of the content in the article is very confusing. So, the paper needs reformulation in order to be clearer and to present a scientific format. Among them, the problems are mainly focused on the introduction of research topics and the description of the Figures & Tables. Here are my specific comments below:

Text content:

  1. The abstract is not well-written thus should be polished.

Answer: The authors rewrote the abstract in a better way:

The formation of heat stable salts (HSS) during the natural gas sweetening process by amine solvent causes many problems such as corrosion, foaming, capacity reduction, and amine loss. A modelling study was carried out for the removal of HSS ions from amine solution using nanofiltration (NF) membrane process that ensures the reuse of amine solution for gas sweetening. This model studies the physics of the nanofiltration process by adjusting and investigating pore radius, the effects of membrane charge, and other membrane characteristics. In this paper, the performance of the ternary ions was investigated during the removal process from methyl di-ethanol amine solution by the nanofiltration membrane process. Correlation between feed concentration and permeate concentration, using experimental results with mathematical correlation as Ci,p = f (Ci,f) was used in modeling. The results showed that the calculated data from the model provided a good agreement with experimental results (R2=0.90-0.75). Also, the effect of operating conditions (including feed pressure and feed flow rate on ions rejection and recovery ratio across the flat-sheet membrane) was studied. The results showed that the recovery and rejection ratios of the NF membrane depend on the driving pressure across the membrane. While the driving pressure is affected by the feed flow conditions and feed pressure.

  1. Advantages and disadvantages of the proposed model in this field should be better discussed aimed to justify the applied structure.

Answer: Advantages and disadvantages of the proposed model are mentioned in the manuscript:

The DSPM-DE has been widely applied in NF systems for feed solutions with single-species and mixed electrolytes. However, the DSPM-DE model has some limitations and many basic assumptions. These include strong sensitivity to some experimental errors, the requirement of excessive experiments to determine necessary parameters, and questionable underlying assumptions of the mean-field approach. Also, although the DSPE-DE applies several significant ion transport mechanisms, it lacks the key features as an ideal model for performance prediction

  1. The content of each part of the article is more like a splicing and explanation of the experiment, without the slightest logic in the middle.

Answer: In this work, the authors have used a combination of experimental results and modeling to analyze the performance of the nanofiltration process in separating ions from amine solution. Various factors such as diffusion coefficients, mass transfer coefficients, Reynolds numbers and hydraulic pressures were calculated to analyze the results to give us an accurate assessment of the modeling results.

Figures & Tables:

  1. There are too many figures in this article and the typesetting is too messy, which makes it very troublesome to read. I hope that it can be deleted and typeset again.
  2. Please keep the font of the figures’ axis title consistent.
  3. The legend in the Fig. 3 is missing.
  4. Line 289: if there are no special circumstances, please keep the same font before and after.
  5. 11 needs to be redrawn.

Answer: The points mentioned by the reviewer were modified and highlighted in the manuscript.

I am sorry to say that this article is difficult to read. The author needs to rethink how this article should be expressed clearly, and the current version does not meet the requirements of publication.

Answer: If the reviewer's opinion was based on misunderstanding we think that after this clarification, thanks to his suggestion, he can reconsider the publication in this Journal.

Reviewer 2 Report

First of all, thank you for providing a manuscript of your work to the membranes journal. I read your article with interest and I have number of comments that I would like to share with you:

General:

  • Your paper is NOT organized based on the Membranes' journal template. Specifically, you must present materials/methods (or in your case model description) prior to the results rection. You must reorganize your paper. It is very hard to follow your content!

Abstract:

  • The connection between first and second sentences is not clear. You should provide more information before you explain the aim of your study.
  • Abstract, should contain your major findings. What was your major findings in this study?

section 2.6:

  • Your model includes component mass balance, total mass balance, and an equation that correlates component's concentration in the permeate side as a function of the component's concentration in the feed side. As you mentioned in lines 369-371, "Ci,p was calculated from the quadratic function which was obtained by fitting the experimental values of Ci,p versus Ci,f". Of course your model prediction will be in good agreement with the experimental results but as a consequence your model is only useful within the fitted range? I would like you to explain more here.
  • Appendix A: is not readable. You must provide a new version with much better quality.

section 2.5:

  • Figure 3: you presented three sets of data in green, blue and red. But  only two colours are tagged. Why? what are the greens? additionaly, on the same data set (green), you plot the obtained results from the mathematical equations. What caused the sudden drop between the last two points in your curve? explain.

Materials and methods:

  • Lines 415-416: the characteristics of the NF-3 membrane are provided in Table 2 and not 1 as you reported!!
  • Table 2: NF-3 is not the membrane type, is the membrane name.
  • Lines 424-426: compnay information should be provided in more detail, preferably a reference link to the company website. The same in lines 427-428, reference is required.
  • Figure 14: low quality, improve it. Also, it must be explained in text. You can not add a figure with no explanation!

Conclusions:

  • Poor!
  • You must at least provide a summary of your main observations, and add a part on recommendations for further research.

Minors:

  • Line 78: "To the best of knowledge" > "to the best of the authors knowledge"
  • Line 357, Figure 13, caption is not complete!
  • Line 378: Equation 9 must be rewritten.
  • In general, if you have any parameter that contains sub/super script, it must appear properly in text as well.

Author Response

   First of all, thank you for providing a manuscript of your work to the membranes journal. I read your article with interest and I have number of comments that I would like to share with you:

General:

  • Your paper is NOT organized based on the Membranes' journal template. Specifically, you must present materials/methods (or in your case model description) prior to the results rection. You must reorganize your paper. It is very hard to follow your content!

Answer: Thank you for the correction, the authors modified it. 

Abstract:

  • The connection between first and second sentences is not clear. You should provide more information before you explain the aim of your study.
  • Abstract, should contain your major findings. What was your major findings in this study?

Answer: The authors rewrote the abstract in a better way:

The formation of heat stable salts (HSS) during the natural gas sweetening process by amine solvent causes many problems such as corrosion, foaming, capacity reduction, and amine loss. A modelling study was carried out for the removal of HSS ions from amine solution using nanofiltration (NF) membrane process that ensures the reuse of amine solution for gas sweetening. This model studies the physics of the nanofiltration process by adjusting and investigating pore radius, the effects of membrane charge, and other membrane characteristics. In this paper, the performance of the ternary ions was investigated during the removal process from methyl di-ethanol amine solution by the nanofiltration membrane process. Correlation between feed concentration and permeate concentration, using experimental results with mathematical correlation as Ci,p = f (Ci,f) was used in modeling. The results showed that the calculated data from the model provided a good agreement with experimental results (R2=0.90-0.75). Also, the effect of operating conditions (including feed pressure and feed flow rate on ions rejection and recovery ratio across the flat-sheet membrane) was studied. The results showed that the recovery and rejection ratios of the NF membrane depend on the driving pressure across the membrane. While the driving pressure is affected by the feed flow conditions and feed pressure.

section 2.6:

  • Your model includes component mass balance, total mass balance, and an equation that correlates component's concentration in the permeate side as a function of the component's concentration in the feed side. As you mentioned in lines 369-371, "Ci,p was calculated from the quadratic function which was obtained by fitting the experimental values of Ci,p versus Ci,f". Of course your model prediction will be in good agreement with the experimental results but as a consequence your model is only useful within the fitted range? I would like you to explain more here.

Answer: The reviewer pointed out a very good point.  Although the mass transfer equations are general and the correlation equations have dependencies and limitations, there is the same concentration range of HSS ions in the amine solution in the gas sweetening process. Therefore, this model can generally be relied upon to evaluate the removal of ions from the amine solution.

  • Appendix A: is not readable. You must provide a new version with much better quality.

Answer: The authors provided a better version of Appendix A.

section 2.5:

  • Figure 3: you presented three sets of data in green, blue and red. But  only two colours are tagged. Why? what are the greens? additionaly, on the same data set (green), you plot the obtained results from the mathematical equations. What caused the sudden drop between the last two points in your curve? explain.

Answer: Green is the sulfate data that is marked in the new version in the Figure. Figure 3 (figure 5  in the new version)  is the comparison of the experimental data with the obtained results of the mathematical equations reported for acetate, formate, and sulfate. This sudden drop in the model result for sulfate is related to equation errors.

Materials and methods:

  • Lines 415-416: the characteristics of the NF-3 membrane are provided in Table 2 and not 1 as you reported!!
  • Table 2: NF-3 is not the membrane type, is the membrane name.
  • Lines 424-426: compnay information should be provided in more detail, preferably a reference link to the company website. The same in lines 427-428, reference is required.
  • Figure 14: low quality, improve it. Also, it must be explained in text. You can not add a figure with no explanation!

Answer: Thank you for the correction, the authors modified them. 

Conclusions:

  • Poor!
  • You must at least provide a summary of your main observations, and add a part on recommendations for further research.

Answer: the authors rewrote the conclusion:

“In this study, nanofiltration tests have been developed by mass transfer balance on the flat-sheet module. The effects of different parameters including feed pressure and flow rate on the rejection of ions and the recovery ratio of the NF membrane have been studied. The feed concentration, permeate concentration, and concentration at the membrane surface along flow direction have also been provided at different feed flow rates and feed pressure. The permeate concentrations were expressed as a function of feed concentrations using experimental results for each ion in the modelling study. The obtained data was examined by comparing the DSPM-DE equations with experimental data to remove the ions of 45 wt.% MDEA solution. A good agreement was observed to predict the performance of nanofiltration processes in large-scale and for a wide range of operating conditions. This work indicates that the performance of NF membranes for the removal of ions from MDEA solution can be well predicted by DSPM-DE modal. This model can be used as a reliable method to simulate the performance of NF in large-scale systems and for a variety of aqueous solutions in a wide range of operating conditions and reduce dependence on experimental data.

The results obtained showed that the concentration of ions in the feed side and in permeate side increased with increasing pressure and/or decreasing the feed flow rate along the flow direction. The results have shown that the rejection ratio of the NF membrane depends on concentration polarization and the driving pressure across the membrane. The recovery ratio increased with an increase in the inlet feed pressure due to the increase of solvent flux through the membrane. “

Minors:

  • Line 78: "To the best of knowledge" > "to the best of the authors knowledge"
  • Line 357, Figure 13, caption is not complete!
  • Line 378: Equation 9 must be rewritten.
  • In general, if you have any parameter that contains sub/super script, it must appear properly in text as well.

Answer: Thank you for the correction, the authors modified them. 

Reviewer 3 Report

The manuscript entitled ” Modeling of Nanofiltration process using DSPM-DE model for purification of amine solution” by Ghorbani et.al. present the findings in modelling NF process. Although the study shows interesting results, my recommendation is that cannot be accepted for publication in its present form.

The structure of the manuscript is rather unconventional and difficult to follow (eg materials and methods are placed above the conclusions section, figures and tables are placed in a separate section).

The experimental part is not properly presented (e.g. feed composition is missing) as well as the duration of experiments. To my understanding the concentrate is returned to the feed tank, this will have an effect in the composition over time, how was this taken into account?  

Is the experimental procedure described identical of this in reference 47?  If yes, the authors don’t need to write it in detail, they can simply state “as was performed in previous study”.

Additionally more information is needed in relation to the ion determination method, deviation of results as well as the repeatability of results.

The manuscript need to be enriched in references (e.g. Table 1, which also needs revision).

My opinion is that the text needs to reconstructed in a major scale in order to become more comprehensible and easier for the reader to follow.  

Author Response

The manuscript entitled ” Modeling of Nanofiltration process using DSPM-DE model for purification of amine solution” by Ghorbani et.al. present the findings in modelling NF process. Although the study shows interesting results, my recommendation is that cannot be accepted for publication in its present form.

Answer: If the reviewer's opinion was based on misunderstanding we think that after this clarification, thanks to his suggestion, he can reconsider the publication in this Journal.

The structure of the manuscript is rather unconventional and difficult to follow (eg materials and methods are placed above the conclusions section, figures and tables are placed in a separate section).

Answer: The authors modified the structure of the manuscript.

The experimental part is not properly presented (e.g. feed composition is missing) as well as the duration of experiments. To my understanding the concentrate is returned to the feed tank, this will have an effect in the composition over time, how was this taken into account?  

Is the experimental procedure described identical of this in reference 47?  If yes, the authors don’t need to write it in detail, they can simply state “as was performed in previous study”.

Additionally more information is needed in relation to the ion determination method, deviation of results as well as the repeatability of results.

Answer: In the experimental section of this work, the authors examined the effect of ion concentrations for ternary salts in amine solution on separation (as shown in Figure 3). Also in the materials and methods section, the feed compositions were mentioned:

“The feed was a ternary mixture of C2H3NaO2, CH2O2 and Na2SO4 in 45 wt. % MDEA solution, which were purchased from Ghatran Shimi Tajhiz, Iran.”

“In order to keep the feed concentration constant, the permeate samples were returned to the feed tank after each test”. It is mentioned in the manuscript and highlighted.

the ion determination method was mentioned by the authors in the manuscript:

“The concentration of ions in permeate was measured by an ion chromatography instrument (I.C., metrosep a SUPP 5-250).”

The manuscript need to be enriched in references (e.g. Table 1, which also needs revision).

Answer: Thank you for the correction, the authors modified it. 

My opinion is that the text needs to reconstructed in a major scale in order to become more comprehensible and easier for the reader to follow.  

Answer: The authors modified the structure of the manuscript.

Round 2

Reviewer 1 Report

This version is better but there are still some errors in format. Have you carefully check the manuscript before submit it? I think the authors should be more careful. 

Author Response

This version is better but there are still some errors in format. Have you carefully check the manuscript before submit it? I think the authors should be more careful. 

Answer: The authors modified the errors in a format based on Membranes Journal. Also, they reviewed the manuscript and highlighted its changes.

Reviewer 2 Report

In general, all equations, and the appendix must be improved in terms of visual quality and readability.

Author Response

In general, all equations, and the appendix must be improved in terms of visual quality and readability.

Answer: All equations and appendix A were improved at this version. Actually, the authors rewrote equations through the equation option of Word.

Reviewer 3 Report

The authors failed to address the comments made during the 1st review round. The revised manuscript appears to be compiled in a hurry in a non consistent and methodic manner. 

My suggestion is the authors to spend the adequate amount of time to improve the manuscript. 

I want to emphasize in the parameters used in Table 1. Authors need to review it in a proper manner and also include the relevant references. 

The manuscript at its present form cannot be accepted for publication. 

Me recommendation is major revision  

Author Response

The authors failed to address the comments made during the 1st review round. The revised manuscript appears to be compiled in a hurry in a non consistent and methodic manner. 

My suggestion is the authors to spend the adequate amount of time to improve the manuscript. 

I want to emphasize in the parameters used in Table 1. Authors need to review it in a proper manner and also include the relevant references. 

Answer: The authors reviewed the manuscript carefully and highlighted new changes. 

The parameters in Table 1 are the characteristics of the NF-3 membrane which were provided by manufacturers  (Sepro Co., USA). Also, the relevant reference was added to the new version of the manuscript ( Environ Sci Pollut Res Int 2015, 22, (8), 6010-23).

Round 3

Reviewer 1 Report

I strongly suggest the authors read some newly published papers and learn from them about the presentation of the results. 
This is the second round of revisement but there are still some problems. For example, the Table 1 should use three-line table and Figure 1 is too simple and not attractive. Try to use some color in you figure. Besides, cite some new papers in your field would be helpful, such as Water Research (2020): 115930 and Chemical Engineering Journal 379 (2020): 122351. Besides, the authors should be careful when write a paper and make sure every word in your paper is right. H2S should be H2S and so on. This kind of issue will consume the patience of your readers and even make them lose confidence in you results.

Author Response

Answer: The authors revised Table 1 and Figure 1. Some new papers (Journal of Membrane Science 2021, 620, 118973, Journal of Membrane Science 2021, 620, 118809, Membranes 2021, 11, (2), 130) were cited in the manuscript. Also, Typographical mistakes in the manuscript were corrected.

The new changes were highlighted in purple. 

Reviewer 3 Report

Unfortunately the manuscript is not improved in such a scale to become appropriate for publication.

Also It is apparent that my comment regarding Table 1 refers to the Table 1 in line 276 "Table 1. Model parameters and physical properties of ions at 35°C." which all the calculations are based upon.

Author Response

Answer: The table title for "Model parameters and physical properties of ions at 35°C" was modified and renamed Table 2. In this work as also mentioned in the manuscript in Section 3.3, the bulk diffusivity of acetate, formate, and sulfate in 45 wt.% MDEA was calculated by molecular dynamic simulation. The mass transfer coefficients and solute radius of ions were estimated by 14-15 eqs. and 19 eq., respectively. Also, feed flow rate,  cross flow velocity, and temperature in the model were determined based on experimental conditions. These parameters were used to simulate the DSPM-DE model. The other mentioned parameters were referenced in the manuscript.

The new changes were highlighted in purple.